# CODE2SEQ: GENERATING SEQUENCES FROM STRUCTURED REPRESENTATIONS OF CODE

**Uri Alon**
Technion
urialon@cs.technion.ac.il

**Shaked Brody**
Technion
shakedbr@cs.technion.ac.il

**Omer Levy**
Facebook AI Research
omerlevy@gmail.com

**Eran Yahav**
Technion
yahave@cs.technion.ac.il

## ABSTRACT

The ability to generate natural language sequences from source code snippets has a variety of applications such as code summarization, documentation, and retrieval. Sequence-to-sequence (seq2seq) models, adopted from neural machine translation (NMT), have achieved state-of-the-art performance on these tasks by treating source code as a sequence of tokens. We present CODE2SEQ: an alternative approach that leverages the syntactic structure of programming languages to better encode source code. Our model represents a code snippet as the set of compositional paths in its abstract syntax tree (AST) and uses attention to select the relevant paths while decoding. We demonstrate the effectiveness of our approach for two tasks, two programming languages, and four datasets of up to 16M examples. Our model significantly outperforms previous models that were specifically designed for programming languages, as well as state-of-the-art NMT models. An online demo of our model is available at http://code2seq.org. Our code, data and trained models are available at http://github.com/tech-srl/code2seq.

## 1 INTRODUCTION

Modeling the relation between source code and natural language can be used for automatic code summarization (Allamanis et al., 2016), documentation (Iyer et al., 2016), retrieval (Allamanis et al., 2015b), and even generation (Balog et al., 2017; Rabinovich et al., 2017; Yin and Neubig, 2017; Devlin et al., 2017; Murali et al., 2017; Brockschmidt et al., 2019). In this work, we consider the general problem of generating a natural language sequence from a given snippet of source code.

A direct approach is to frame the problem as a machine translation problem, where the source sentence is the sequence of tokens in the code and the target sentence is a corresponding natural language sequence. This approach allows one to apply state-of-the-art neural machine translation (NMT) models from the sequence-to-sequence (seq2seq) paradigm (Sutskever et al., 2014; Cho et al., 2014; Bahdanau et al., 2014; Luong et al., 2015; Vaswani et al., 2017), yielding state-of-the-art performance on various code captioning and documentation benchmarks (Iyer et al., 2016; Allamanis et al., 2016; Loyola et al., 2017) despite having extremely long source sequences.

We present an alternative approach for encoding source code that leverages the syntactic structure of programming languages: CODE2SEQ. We represent a given code snippet as a set of compositional paths over its abstract syntax tree (AST), where each path is compressed to a fixed-length vector using LSTMs (Hochreiter and Schmidhuber, 1997). During decoding, CODE2SEQ attends over a different weighted average of the path-vectors to produce each output token, much like NMT models attend over token representations in the source sentence.

We show the effectiveness of our code2seq model on two tasks: (1) code summarization (Figure 1a), where we predict a Java method's name given its body, and (2) code captioning (Figure 1b), where we predict a natural language sentence that describes a given C# snippet.

**Code summarization in Java:**

```
public boolean ______ ① (Set<String> set,
                                    String value) {
    for (String entry) : set) {
②
        if (entry.equalsIgnoreCase(value))
③          return true;
    }
    }
    return false;
}
```

contains① ignore② case③

(a)

**Code captioning in C#:**

```
void Main() {
    string text = File.ReadAllText(@"T:\File1.txt");
    int num = 0;
                    ①          ②
    text = (Regex.Replace(text, "map", delegate (Match m) {
        return "map" + num++;
    }));
    File.WriteAllText(@"T:\File1.txt", text);
}
```

replace① a string② in a text③ file③

(b)

Figure 1: Example of (a) code summarization of a Java code snippet, and (b) code captioning of a C# code snippet, along with the predictions produced by our models. The highlighted paths in each example are the top-attended paths in each decoding step. Because of space limitations we included only the top-attended path for each decoding step, but hundreds of paths are attended at each step. Additional examples are presented in Appendix B and Appendix C.

On both tasks, our CODE2SEQ model outperforms models that were explicitly designed for code, such as the model of Allamanis et al. (2016) and CodeNN (Iyer et al., 2016), as well as TreeLSTMs (Tai et al., 2015) and state-of-the-art NMT models (Luong et al., 2015; Vaswani et al., 2017). To examine the importance of each component of the model, we conduct a thorough ablation study. In particular, we show the importance of structural encoding of code, by showing how our model yields a significant improvement over an ablation that uses only token-level information without syntactic paths. To the best of our knowledge, this is the first work to directly use paths in the abstract syntax tree for end-to-end generation of sequences.

## 2 REPRESENTING CODE AS AST PATHS

An Abstract Syntax Tree (AST) uniquely represents a source code snippet in a given language and grammar. The leaves of the tree are called *terminals*, and usually refer to user-defined values which represent identifiers and names from the code. The non-leaf nodes are called *nonterminals* and represent a restricted set of structures in the language, e.g., loops, expressions, and variable declarations. For example, Figure 2c shows a partial AST for the code snippet of Figure 2a. Names (such as `num`) and types (such as `int`) are represented as values of terminals; syntactic structures such as variable declaration (`VarDec`) and a do-while loop (`DoStmt`) are represented as nonterminals.

Given the AST of a code snippet, we consider all pairwise paths between terminals, and represent them as sequences of terminal and nonterminal nodes. We then use these paths with their terminals' values to represent the code snippet itself. For example, consider the two Java methods of Figure 2. Both of these methods count occurrences of a character in a string. They have exactly the same functionality, although a different implementation, and therefore different surface forms. If these snippets are encoded as sequences of tokens, the recurring patterns that suggest the common method name might be overlooked. However, a structural observation reveals syntactic paths that are common to both methods, and differ only in a single node of a `Do`-while statement versus a `For` statement. This example shows the effectiveness of a syntactic encoding of code. Such an encoder can generalize much better to unseen examples because the AST normalizes a lot of the surface form variance. Since our encoding is compositional, the encoder can generalize even if the paths are not identical (e.g., a `For` node in one path and a `While` in the other).

Since a code snippet can contain an arbitrary number of such paths, we sample $k$ paths as the representation of the code snippet. To avoid bias, $k$ new paths are sampled afresh in every training iteration. In Section 5 we show that this runtime-sampling provides regularization and improves results compared to sampling the same $k$ paths for each example in advance.

Formally, we use $\mathcal{C}$ to denote a given snippet of code. Every training iteration, $k$ pairs of terminals are uniformly sampled from within the AST of $\mathcal{C}$. Each pair of terminals $\left(v_1^i, v_{l_i}^i\right)$ implies a single

```
int countOccurrences(String str, char ch) {
    int num = 0;
    int index = -1;
    do {
        index = str.indexOf(ch, index + 1);
        if (index >= 0) {
            num++;
        }
    } while (index >= 0);
    return num;
}
```

(a)

```
int countOccurrences(String source, char value) {
    int count = 0;
    for (int i = 0; i < source.length(); i++) {
        if (source.charAt(i) == value) {
            count++;
        }
    }
    return count;
}
```

(b)

(c)

(d)

Figure 2: An example of two Java methods that have exactly the same functionality. Although these methods have different *sequential* (token-based) representations, repeating paths, which might differ in only a single node (a `ForStmt` node instead of a `Do`-while node), will be revealed if we consider syntactic patterns.

path between them: $v_1^i v_2^i ... v_{l_i}^i$. Finally, the input code example is represented as a set of these $k$ random AST paths: $\left\{ \left( v_1^1 v_2^1 ... v_{l_1}^1 \right), ..., \left( v_1^k v_2^k ... v_{l_k}^k \right) \right\}$, where $l_j$ is the length of the $j$th path.

## 3 MODEL ARCHITECTURE

Our model follows the standard encoder-decoder architecture for NMT (Section 3.1), with the significant difference that *the encoder does not read the input as a flat sequence of tokens*. Instead, the encoder creates a vector representation for each AST path separately (Section 3.2). The decoder then attends over the encoded AST paths (rather than the encoded tokens) while generating the target sequence. Our model is illustrated in Figure 3.

### 3.1 ENCODER-DECODER FRAMEWORK

Contemporary NMT models are largely based on an encoder-decoder architecture (Cho et al., 2014; Sutskever et al., 2014; Luong et al., 2015; Bahdanau et al., 2014), where the encoder maps an input sequence of tokens $x = (x_1, ..., x_n)$ to a sequence of continuous representations $z = (z_1, ..., z_n)$. Given $z$, the decoder then generates a sequence of output tokens $y = (y_1, ..., y_m)$ one token at a time, hence modeling the conditional probability: $p(y_1, ..., y_m | x_1, ..., x_n)$.

At each decoding step, the probability of the next target token depends on the previously generated token, and can therefore be factorized as:

$$p(y_1, ..., y_m | x_1, ..., x_n) = \prod_{j=1}^{m} p(y_j | y_{<j}, z_1, ..., z_n)$$

In attention-based models, at each time step $t$ in the decoding phase, a context vector $c_t$ is computed by attending over the elements in $z$ using the decoding state $h_t$, typically computed by an LSTM.

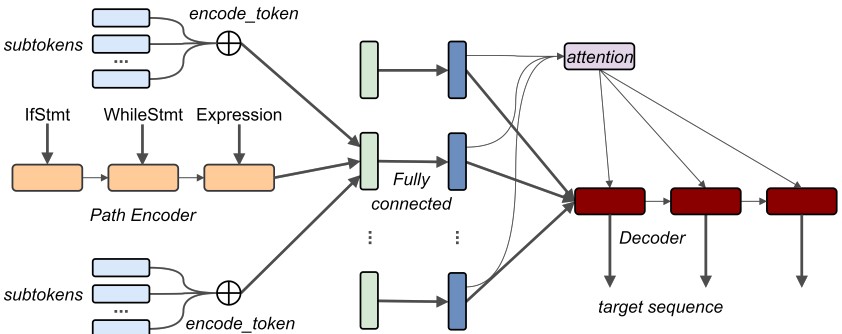

Figure 3: Our model encodes each AST path with its values as a vector, and uses the average of all of the $k$ paths as the decoder's start state. The decoder generates an output sequence while attending over the $k$ encoded paths.

$$\boldsymbol{\alpha}^t = softmax\left(h_t W_a \boldsymbol{z}\right) \qquad c_t = \sum_i^n \alpha_i^t z_i$$

The context vector $c_t$ and the decoding state $h_t$ are then combined to predict the current target token $y_t$. Previous work differs in the way the context vector is computed and in the way it is combined with the current decoding state. A standard approach (Luong et al., 2015) is to pass $c_t$ and $h_t$ through a multi-layer perceptron (MLP) and then predict the probability of the next token using softmax:

$$p\left(y_t | y_{<t}, z_1, ..., z_n\right) = softmax\left(W_s tanh\left(W_c\left[c_t; h_t\right]\right)\right)$$

### 3.2 AST Encoder

Given a set of AST paths $\{x_1, ..., x_k\}$, our goal is to create a vector representation $z_i$ for each path $x_i = v_1^i v_2^i ... v_{l_i}^i$. We represent each path separately using a bi-directional LSTM to encode the path, and sub-token embeddings to capture the compositional nature of the terminals' values (the tokens).

**Path Representation**   Each AST path is composed of nodes and their child indices from a limited vocabulary of up to 364 symbols. We represent each node using a learned embedding matrix $E^{nodes}$ and then encode the entire sequence using the final states of a bi-directional LSTM:

$$h_1, ..., h_l = LSTM(E_{v_1}^{nodes}, ..., E_{v_l}^{nodes})$$
$$encode\_path(v_1...v_l) = [\overrightarrow{h_l}; \overleftarrow{h_1}]$$

**Token Representation**   The first and last node of an AST path are terminals whose values are tokens in the code. Following Allamanis et al. (2015a; 2016), we split code tokens into *sub*tokens; for example, a token with the value `ArrayList` will be decomposed into `Array` and `List`. This is somewhat analogous to byte-pair encoding in NMT (Sennrich et al., 2016), although in the case of programming languages, coding conventions such as camel notation provide us with an explicit partition of each token. We use a learned embedding matrix $E^{subtokens}$ to represent each subtoken, and then sum the subtoken vectors to represent the full token:

$$encode\_token(w) = \sum_{s \in split(w)} E_s^{subtokens}$$

The LSTM decoder may also predict subtokens at each step (e.g. when generating method names), although the decoder's subtoken embedding matrix will be different.

**Combined Representation**   To represent the path $x = v_1...v_l$, we concatenate the path's representation with the token representations of each terminal node, and apply a fully-connected layer:

$$z = tanh\left(W_{in}\left[encode\_path(v_1...v_l); encode\_token(value(v_1)); encode\_token(value(v_l))\right]\right)$$

where $value$ is the mapping of a terminal node to its associated value, and $W_{in}$ is a $(2d_{path} + 2d_{token}) \times d_{hidden}$ matrix.

**Decoder Start State** To provide the decoder with an initial state, we average the combined representations of *all* the $k$ paths in the given example:

$$h_0 = \frac{1}{k} \sum_{i=1}^{k} z_i$$

Unlike typical encoder-decoder models, the order of the input random paths is not taken into account. Each path is encoded separately and the combined representations are aggregated with mean pooling to initialize the decoder's state. This represents the given source code as a *set* of random paths.

**Attention** Finally, the decoder generates the output sequence while *attending over all of the combined representations* $z_1, ... z_k$, similarly to the way that seq2seq models attend over the source symbols. The attention mechanism is used to dynamically select the distribution over these $k$ combined representations while decoding, just as an NMT model would attend over the encoded source tokens.

## 4 EXPERIMENTS

We evaluate our model on two code-to-sequence tasks: summarization (Section 4.1), in which we predict Java methods' names from their bodies, and captioning (Section 4.2), where we generate natural language descriptions of C# code snippets. Although out of the focus of this work, in Section 4.3 we show that our model also generates Javadocs more accurately than an existing work. We thus demonstrate that our approach can produce both method names and natural language outputs, and can encode a code snippet in any language for which an AST can be constructed (i.e., a parser exists).

**Setup** The values of all of the parameters are initialized using the initialization heuristic of Glorot and Bengio (2010). We optimize the cross-entropy loss (Rubinstein, 1999; 2001) with a Nesterov momentum (Nesterov, 1983) of $0.95$ and an initial learning rate of $0.01$, decayed by a factor of $0.95$ every epoch. For the Code Summarization task, we apply dropout (Srivastava et al., 2014) of $0.25$ on the input vectors $x_j$, and $0.7$ for the Code Captioning task because of the smaller number of examples in the C# dataset. We apply a recurrent dropout of $0.5$ on the LSTM that encodes the AST paths. We used $d_{tokens} = d_{nodes} = d_{hidden} = d_{target} = 128$. For the Code Summarization task, each LSTM that encodes the AST paths had 128 units and the decoder LSTM had 320 units. For the Code Captioning task, to support the longer target sequences, each encoder LSTM had 256 units and the decoder was of size $512$.

**Choice of $k$** We experimented with different values of $k$, the number of sampled paths from each example (which we set to 200 in the final models). Lower values than $k = 100$ showed worse results, and increasing to $k > 300$ did not result in consistent improvement. In practice, $k = 200$ was found to be a reasonable sweet spot between capturing enough information while keeping training feasible in the GPU's memory. Additionally, since the average number of paths in our Java-large training set is 220 paths per example, a number as high as 200 is beneficial for some large methods.

### 4.1 CODE SUMMARIZATION

In this task, we predict a Java method's name given its body. As was previously observed (Allamanis et al., 2016; Alon et al., 2019), this is a good benchmark because a method name in open-source Java projects tends to be succinct and precise, and a method body is often a complete logical unit. We predict the target method name as a sequence of sub-tokens, e.g., `setMaxConnectionsPerServer` is predicted as the sequence "set max connections per server". The target sequence length is about 3 on average. We adopt the measure used by Allamanis et al. (2016) and Alon et al. (2019), who measured precision, recall, and F1 score over the target sequence, case insensitive.

**Data** We experiment with this task across three datsets. In these datasets, we always train across multiple projects and predict on distinct projects:

*Java-small* – Contains 11 relatively large Java projects, originally used for 11 distinct models for training and predicting within the scope of the same project (Allamanis et al., 2016). We use the same data, but *train and predict across projects*: we took 9 projects for training, 1 project for validation and 1 project as our test set. This dataset contains about $700K$ examples.

Table 1: Our model significantly outperforms previous PL-oriented and NMT models. Another visualization can be found in Appendix E.

| Model | Java-small | | | Java-med | | | Java-large | | |
|---|---|---|---|---|---|---|---|---|---|
| | Prec | Rec | F1 | Prec | Rec | F1 | Prec | Rec | F1 |
| ConvAttention (Allamanis et al., 2016) | 50.25 | 24.62 | 33.05 | 60.82 | 26.75 | 37.16 | 60.71 | 27.60 | 37.95 |
| Paths+CRFs (Alon et al., 2018) | 8.39 | 5.63 | 6.74 | 32.56 | 20.37 | 25.06 | 32.56 | 20.37 | 25.06 |
| code2vec (Alon et al., 2019) | 18.51 | 18.74 | 18.62 | 38.12 | 28.31 | 32.49 | 48.15 | 38.40 | 42.73 |
| 2-layer BiLSTM (no token splitting) | 32.40 | 20.40 | 25.03 | 48.37 | 30.29 | 37.25 | 58.02 | 37.73 | 45.73 |
| 2-layer BiLSTM | 42.63 | 29.97 | 35.20 | 55.15 | 41.75 | 47.52 | 63.53 | 48.77 | 55.18 |
| TreeLSTM (Tai et al., 2015) | 40.02 | 31.84 | 35.46 | 53.07 | 41.69 | 46.69 | 60.34 | 48.27 | 53.63 |
| Transformer (Vaswani et al., 2017) | 38.13 | 26.70 | 31.41 | 50.11 | 35.01 | 41.22 | 59.13 | 40.58 | 48.13 |
| code2seq | **50.64** | **37.40** | **43.02** | **61.24** | **47.07** | **53.23** | **64.03** | **55.02** | **59.19** |
| Absolute gain over BiLSTM | +8.01 | +7.43 | +7.82 | +6.09 | +5.32 | +5.71 | +0.50 | +6.25 | +4.01 |

*Java-med* – A new dataset of the 1000 top-starred Java projects from GitHub. We randomly select 800 projects for training, 100 for validation and 100 for testing. This dataset contains about 4M examples and we make it publicly available.

*Java-large* – A new dataset of the 9500 top-starred Java projects from GitHub that were created since January 2007. We randomly select 9000 projects for training, 250 for validation and 300 for testing. This dataset contains about $16M$ examples and we make it publicly available.

More statistics of our datasets can be found in Appendix A.

**Baselines** We re-trained all of the baselines on all of the datasets of this task using the original implementations of the authors. We compare CODE2SEQ to the following baselines: Allamanis et al. (2016), who used a convolutional attention network to predict method names; syntactic paths with Conditional Random Fields (CRFs) (Alon et al., 2018); code2vec (Alon et al., 2019); and a TreeLSTM (Tai et al., 2015) encoder with an LSTM decoder and attention on the input sub-trees. Additionally, we compared to three NMT baselines that read the input source code as a stream of tokens: 2-layer bidirectional encoder-decoder LSTMs (split tokens and full tokens) with global attention (Luong et al., 2015), and the Transformer (Vaswani et al., 2017), which achieved state-of-the-art results for translation tasks.

We put significant effort into strengthening the NMT baselines in order to provide a fair comparison: (1) we split tokens to subtokens, as in our model (e.g., `HashSet → Hash Set`) – this was shown to improve the results by about 10 F1 points (Table 1); (2) we deliberately kept the original casing of the source tokens since we found it to improve their results; and (3) during inference, we replaced generated UNK tokens with the source tokens that were given the highest attention. For the 2-layer BiLSTM we used embeddings of size 512, an encoder and a decoder of 512 units each, and the default hyperparameters of OpenNMT (Klein et al., 2017). For the Transformer, we used their original hyperparameters (Vaswani et al., 2017). This resulted in a Transformer model with $169M$ parameters and a BiLSTM model with $134M$ parameters, while our code2seq model had only 37M.[1]

**Performance** Table 1 shows the results for the code summarization task. Our model significantly outperforms the baselines in both precision and recall across all three datasets, demonstrating that there is added value in leveraging ASTs to encode source code. Our model improves over the best baselines, BiLSTM with split tokens, by between $4$ to $8$ F1 points on all benchmarks. BiLSTM with split tokens consistently scored about 10 F1 points more than BiLSTM with full tokens, and for this reason we included only the split token Transformer and TreeLSTM baselines. Our model outperforms ConvAttention (Allamanis et al., 2016), which was designed specifically for this task; Paths+CRFs (Alon et al., 2018), which used syntactic features; and TreeLSTMs. Although TreeLSTMs also leverage syntax, we hypothesize that our syntactic paths capture long distance relationships while TreeLSTMs capture mostly local properties. An additional comparison to code2vec on the code2vec dataset can be found in Appendix A. Examples for predictions made by our model and each of the baselines can be found in Appendix C and at `http://code2seq.org`.

---

[1]We also trained versions of the NMT baselines in which we down-matched the sizes and number of parameters to our model. These baselines seemed to benefit from more parameters, so the results reported here are for the versions that had many more parameters than our model.

Fernandes et al. (2019) encoded code using Graph Neural Networks (GNN), and reported lower performance than our model on Java-large without specifying the exact F1 score. They report slightly higher results than us on Java-small only by extending their GNN encoder with a subtoken-LSTM (BiLSTM+GNN→ LSTM); by extending the Transformer with GNN (SELFATT+GNN→SELFATT); or by extending their LSTM decoder with a pointer network (GNN→LSTM+POINTER). All these extensions can be incorporated into our model as well.

**Data Efficiency** ConvAttention (Allamanis et al., 2016) performed even better than the Transformer on the Java-small dataset, but could not scale and leverage the larger datasets. Paths+CRFs showed very poor results on the Java-small dataset, which is expected due to the sparse nature of their paths and the CRF model. When compared to the best among the baselines (BiLSTM with split tokens), our model achieves a relative improvement of 7.3% on Java-large, but as the dataset becomes smaller, the larger the relative difference becomes: 13% on Java-med and 22% on Java-small; when compared to the Transformer, the relative improvement is 23% on Java-large and 37% on Java-small. These results show the data efficiency of our architecture: while the data-hungry NMT baselines *require* large datasets, our model can leverage both small and large datasets.

**Sensitivity to input length** We examined how the performance of each model changes as the size of the test method grows. As shown in Figure 4, our model is superior to all examined baselines across all code lengths. All models give their best results for short snippets of code, i.e., less than 3 lines. As the size of the input code increases, all examined models show a natural descent, and show stable results for lengths of 9 and above.

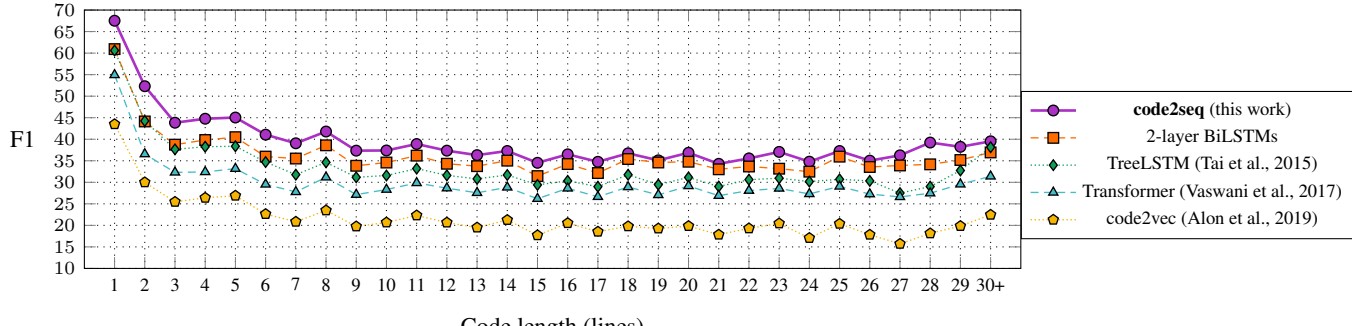

Figure 4: F1 score compared to the length of the input code. This experiment was performed for the code summarization task on the Java-med test set. All examples having more than 30 lines were counted as having 30 lines.

## 4.2 CODE CAPTIONING

For this task we consider predicting a full natural language sentence given a short C# code snippet. We used the dataset of CodeNN (Iyer et al., 2016), which consists of 66,015 pairs of questions and answers from StackOverflow. They used a semi-supervised classifier to filter irrelevant examples and asked human annotators to provide two additional titles for the examples in the test set, making a total of three reference titles for each code snippet. The target sequence length in this task is about 10 on average. This dataset is especially challenging as it is orders of magnitude smaller than the code summarization datasets. Additionally, StackOverflow code snippets are typically short, incomplete at times, and aim to provide an answer to a very specific question. We evaluated using BLEU score with smoothing, using the same evaluation scripts as Iyer et al. (2016).

**Baselines** We present results compared to CodeNN, TreeLSTMs with attention, 2-layer bidirectional LSTMs with attention, and the Transformer. As before, we provide a fair comparison by splitting tokens to subtokens, and replacing UNK during inference. We also include numbers from baselines used by Iyer et al. (2016).

**Results** Table 2 summarizes the results for the code captioning task. Our model achieves a BLEU score of 23.04, which improves by 2.51 points (12.2% relative) over CodeNN, whose authors introduced this dataset, and over all the other baselines, including BiLSTMs, TreeLSTMs and the

Table 2: Our model outperforms previous work in the code captioning task. [†]Results previously reported by Iyer et al. (2016), and verified by us. Another visualization can be found in Appendix D.

| Model | BLEU |
|---|---|
| MOSES[†] (Koehn et al., 2007) | 11.57 |
| IR[†] | 13.66 |
| SUM-NN[†] (Rush et al., 2015) | 19.31 |
| 2-layer BiLSTM | 19.78 |
| Transformer (Vaswani et al., 2017) | 19.68 |
| TreeLSTM (Tai et al., 2015) | 20.11 |
| CodeNN[†] (Iyer et al., 2016) | 20.53 |
| code2seq | **23.04** |

Table 3: Variations on the code2seq model, performed on the validation set of Java-med.

| Model | Precision | Recall | F1 | $\Delta$F1 |
|---|---|---|---|---|
| code2seq (original model) | **60.67** | **47.41** | **53.23** | |
| No AST nodes (only tokens) | 55.51 | 43.11 | 48.53 | -4.70 |
| No decoder | 47.99 | 28.96 | 36.12 | -17.11 |
| No token splitting | 48.53 | 34.80 | 40.53 | -12.70 |
| No tokens (only AST nodes) | 33.78 | 21.23 | 26.07 | -27.16 |
| No attention | 57.00 | 41.89 | 48.29 | -4.94 |
| No random (sample $k$ paths in advance) | 59.08 | 44.07 | 50.49 | -2.74 |

Transformer, which achieved slightly lower results than CodeNN. Examples for predictions made by our model and each of the baselines can be found in Appendix F. These results show that when the training examples are short and contain incomplete code snippets, our model generalizes better to unseen examples than a shallow textual token-level approach, thanks to its syntactic representation of the data. Although TreeLSTMs also represent the data syntactically, the TreeLSTM baseline achieved lower scores.

### 4.3 CODE DOCUMENTATION

Although the task of generating code documentation is outside the focus of this work, we performed an additional comparison to Hu et al. (2018). They trained a standard seq2seq model by using the linearized AST as the source sequence and a Javadoc natural language sentence as the target sequence. While they originally report a BLEU score of 38.17, we computed their BLEU score using prediction logs provided us by the authors and obtained a BLEU score of 8.97, which we find more realistic. Training our model on the same dataset as Hu et al., matching LSTM sizes, and using the same script on our predictions yields a BLEU score of 14.53, which is a 62% relative gain over the model of Hu et al. (2018). This shows that our structural approach represents code better than linearizing the AST and learning it as a sequence.

### 5 ABLATION STUDY

To better understand the importance of the different components of our model, we conducted an extensive ablation study. We varied our model in different ways and measured the change in performance. These experiments were performed for the code summarization task, on the validation set of the Java-med dataset. We examined several alternative designs:

1. *No AST nodes* – instead of encoding an AST path using an LSTM, take only the first and last terminal values to construct an input vector

2. *No decoder* – no sequential decoding; instead, predict the target sequence as a single symbol using a single softmax layer.

3. *No token splitting* – no subtoken encoding; instead, embed the full token.

4. *No tokens* – use only the AST nodes without using the values associated with the terminals.

5. *No attention* – decode the target sequence given the initial decoder state, without attention.

6. *No random* – no re-sampling of $k$ paths in each iteration; instead, sample in advance and use the same $k$ paths for each example throughout the training process.

Table 3 shows the results of these alternatives. As seen, *not encoding AST nodes* resulted in a degradation especially in the precision: a decrease of $5.16$ compared to $4.30$ for the recall. It is quite surprising that this ablation was still better than the baselines (Table 1): for example, the Transformer can implicitly capture pairs of tokens using its self-attention mechanism. However, *not all tokens are AST leaves*. By focusing on AST leaves, we increase the focus on named tokens, and effectively ignore functional tokens like brackets, parentheses, semicolons, etc. Transformers can (in theory) capture the same signal, but perhaps they require significantly more layers or a different optimization to actually learn to focus on those particular elements. The AST gives us this information for free without having to spend more transformer layers just to learn it. Additionally, for practical reasons we limited the length of the paths to $9$ . This leads to pairs of leaves that are close in the AST, but not necessarily close in the sequence. In contrast, the Transformer's attention is effectively skewed towards sequential proximity because of the positional embeddings.

Using a single prediction with *no decoder* reduces recall by more than one-third. This shows that the method name prediction task should be addressed as a sequential prediction, despite the methods' relatively short names. Using *no token splitting* or *no tokens* at all drastically reduces the score, showing the significance of encoding both subtokens and syntactic paths. Despite the poor results of *no tokens*, it is still surprising that the model can achieve around half the score of the full model, as using no tokens is equivalent to reasoning about code which has no identifier names, types, APIs, and constant values, which can be very difficult even for a human. The *no attention* experiment shows the contribution of attention in our model, which is very close in its relative value to the contribution of attention in seq2seq models (Luong et al., 2015; Bahdanau et al., 2014). The *no random* experiment shows the positive contribution of sampling $k$ different paths afresh on every training iteration, instead of using the same sample of paths from each example during the entire training. This approach provides data-level regularization that further improves an already powerful model. Another visualization can be found in Appendix D.

## 6    RELATED WORK

The growing availability of open source repositories creates new opportunities for using machine learning to process source code en masse. Several papers model code as a sequence of tokens (Iyer et al., 2016; Allamanis et al., 2016; Loyola et al., 2017), characters (Bielik et al., 2017), and API calls (Raychev et al., 2014). While sometimes obtaining satisfying results, these models treat code as a sequence rather than a tree. This necessitates implicit relearning of the (predefined) syntax of the programming language, wasting resources and reducing accuracy.

Code representation models that use syntactic information have usually been evaluated on relatively easier tasks, which mainly focus on "filling the blanks" in a given program (Alon et al., 2018; Bielik et al., 2016; Raychev et al., 2016; 2015; Allamanis et al., 2018) or semantic classification of code snippets (Alon et al., 2019). Moreover, none of the models that use syntactic relations are compositional, and therefore the number of possible syntactic relations is fixed either before or after training, a process which results in a large RAM and GPU memory consumption. The syntactic paths of Alon et al. (2018; 2019) are represented *monolithically*, and are therefore limited to only a subset of the paths that were observed enough times during training. As a result, *they cannot represent unseen relations*. In contrast, by representing AST paths node-by-node using LSTMs, our model can represent and use *any* syntactic path in any unseen example. Further, our model decodes the output sequence step-by-step while attending over the input paths, and can thus generate unseen sequences, compared to code2vec (Alon et al., 2019), which has a closed vocabulary.

Oda et al. (2015) were the first to generate sequences by leveraging the syntax of code. They performed a line-by-line statistical machine translation (SMT) to translate Python code to pseudo-code. Our tasks are different, and we cannot assume an alignment between elements in the input and the output; our tasks take a whole code snippet as their input, and produce a much shorter sequence as output. Additionally, a conceptual advantage of our model over line-by-line translation is its ability to capture multiline patterns in the source code. These multiline patterns are often very useful for the model and get the most attention (Figure 1a). A recent work (Hu et al., 2018) generates

comments from code. There is a conceptual difference between our approaches: Hu et al. (2018) linearize the AST, and then pass it on to a standard seq2seq model. We present a new model, in which the encoder already assumes that the input is tree-structured. When training our model on their dataset, we improve over their BLEU score by $62\%$ (Section 4.3).

Allamanis et al. (2018) represent code with Gated Graph Neural Networks. Nodes in the graph represent identifiers, and edges represent syntactic and semantic relations in the code such as "ComputedFrom" and "LastWrite". The edges are designed for the semantics of a specific programming language, for a specific task, and require an expert to devise and implement. In contrast, our model has minimal assumptions on the input language and is general enough not to require either expert semantic knowledge or the manual design of features. Our model can therefore be easily implemented for various input languages. Bastings et al. (2017) used graph-convolutional networks for machine translation of natural languages. Piech et al. (2015) encoded code using Tree-RNNs to propagate feedback on student code; and Chen et al. (2018) used Tree-RNNs for a tree-to-tree translation of code into another programming language.

## 7 CONCLUSION

We presented a novel code-to-sequence model which considers the unique syntactic structure of source code with a sequential modeling of natural language. The core idea is to sample paths in the Abstract Syntax Tree of a code snippet, encode these paths with an LSTM, and attend to them while generating the target sequence.

We demonstrate our approach by using it to predict method names across three datasets of varying sizes, predict natural language captions given partial and short code snippets, and to generate method documentation, in two programming languages. Our model performs significantly better than previous programming-language-oriented works and state-of-the-art NMT models applied in our settings.

We believe that the principles presented in this paper can serve as a basis for a wide range of tasks which involve source code and natural language, and can be extended to other kinds of generated outputs. To this end, we make all our code, datasets, and trained models publicly available.

ACKNOWLEDGMENTS

We would like to thank Guy Waldman for developing the code2seq website (`http://code2seq.org`), Srinivasan Iyer for the guidance in using his C# dataset, and Miltiadis Allamanis, Yoav Goldberg, Charles Sutton and the anonymous reviewers for their fruitful comments and suggestions.

The research leading to these results has received funding from the European Union's Seventh Framework Programme (FP7) under grant agreement no. 615688-ERC-COG-PRIME, and the Israeli Science Foundation (ISF) under grant agreement #1319/16.

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

Table 4: Our model significantly outperforms code2vec on the code2vec dataset.

| Model | Our Java-large dataset (same as in Table 1) | | | code2vec dataset (Alon et al., 2019) | | |
|---|---|---|---|---|---|---|
| | Prec | Rec | F1 | Prec | Rec | F1 |
| code2vec (Alon et al., 2019) | 48.2 | 38.4 | 42.7 | 63.1 | 54.4 | 58.4 |
| code2seq (this work) | **64.0** | **55.02** | **59.2** | **70.2** | **63.3** | **66.6** |

Table 5: Statistics of our datasets.

| | Java-small | Java-med | Java-large | C# (Iyer et al., 2016) |
|---|---|---|---|---|
| #projects - training | 10 | 800 | 8999 | - |
| #projects - validation | 1 | 100 | 250 | - |
| #projects - test | 1 | 96 | 307 | - |
| #examples - training | 665,115 | 3,004,536 | 15,344,512 | 52,812 |
| #examples - validation | 23,505 | 410,699 | 320,866 | 6,601 |
| #examples - test | 56,165 | 411,751 | 417,003 | 6,602 |
| Avg. number of paths (training) | 171 | 187 | 220 | 207 |
| Avg. code length - lines (training) | 6.0 | 6.3 | 6.6 | 8.3 |
| Avg. code length - tokens (training) | 60 | 63 | 65 | 38 |
| Avg. code length - subtokenized (training) | 75 | 78 | 80 | 67 |
| Avg. target length (training) | 3 | 3 | 3 | 10 |

## A    ADDITIONAL EVALUATION

**Comparison to code2vec on their dataset**    We perform an additional comparison to code2vec (Alon et al., 2019) on their proposed dataset. As shown in Table 4, code2vec achieves a high F1 score on that dataset. However, our model achieves an even higher F1 score. The poorer performance of code2vec on our dataset is probably due to its always being split to train/validation/test *by project*, whereas the dataset of code2vec is split *by file*. In the code2vec dataset, a file can be in the training set, while another file from the same project can be in the test set. This makes their dataset significantly easier, because method names "leak" to other files in the same project, and there are often duplicates in different files of the same project. This is consistent with Allamanis et al. (2018), who found that splitting by file makes the dataset easier than by project. We decided to take the stricter approach, and not to use their dataset (even though our model achieves better results on it), in order to make all of our comparisons on split-by-project datasets.

**Data statistics**    Table 5 shows some statistics of our used datasets.

## B    CODE CAPTIONING EXAMPLES

Figure 6 contains examples from our test set for the code captioning task in C#, along with the prediction of our model and each of the baselines.

Figure 5 shows a timestep-by-timestep example, with the symbol decoded at each timestep and the top-attended path at that step. The width of the path is proportional to the attention it was given by the model (because of space limitations we included only the top-attended path for each decoding step, but hundreds of paths are attended at each step).

## C    CODE SUMMARIZATION EXAMPLES

Figure 8 contains examples from our test set for the code summarization task in C#, along with the prediction of our model and each of the baselines. The presented predictions are made by models that were trained on the same Java-large dataset.

## D    CODE CAPTIONING RESULTS

Figure 9 shows a bar chart of the BLEU score of our model and the baselines, in the code captioning task (predicting natural language descriptions for C# code snippets). The numbers are the same as in Table 2.

# E  CODE SUMMARIZATION RESULTS

Figure 10 shows a bar chart of the F1 score of our model and the baselines, in the code summarization task (predicting method names in Java). The numbers are the F1 columns from Table 1.

# F  ABLATION STUDY RESULTS

Figure 11 shows a bar chart of the relative decrease in precision and recall for each of the ablations described in Section 5 and presented in Table 3.

Figure 5: Example of code captioning for a C# code snippet from our test set. The text boxes at the bottom of each figure are the predictions produced by our model at each decoding step. The highlighted paths in each figure are the top-attended paths in each decoding step, and their widths are proportional to their attention weight (because of space limitations we included only the top-attended path for each decoding step, but hundreds of paths are attended at each step).

```
TreeView myTreeView = new TreeView();
myTreeView.Nodes.Clear();
foreach (string parentText in xml.parent)
{
  TreeNode parent = new TreeNode();
  parent.Text = parentText;
  myTreeView.Nodes.Add(treeNodeDivisions);

  foreach (string childText in xml.child)
  {
    TreeNode child = new TreeNode();
    child.Text = childText;
    parent.Nodes.Add(child);
  }
}
```

| Model | Prediction |
|---|---|
| MOSES[†] (Koehn et al., 2007) | How can TreeView TreeView a TreeView nodes from XML parentText string to a treeview node from a TreeView parentText of a tree treeNodeDivisions from to child childText XML node of MDI child childText created in a tree nodes in |
| IR[†] | How to set the name of a tabPage progragmatically |
| SUM-NN[†] (Rush et al., 2015) | how to get data from xml file in c# |
| 2-layer BiLSTM | how to add child nodes to treeview |
| Transformer (Vaswani et al., 2017) | how to add child node in treeview in c # |
| TreeLSTM (Tai et al., 2015) | how to get the value of a node in xml |
| CodeNN[†] (Iyer et al., 2016) | How to get all child nodes in TreeView ? |
| code2seq (this work) | add a child node to a treeview in c # |

```
var excel = new ExcelQueryFactory("excelFileName");
var firstRow = excel.Worksheet().First();
var companyName = firstRow["CompanyName"];
```

| Model | Prediction |
|---|---|
| MOSES[†] (Koehn et al., 2007) | How into string based on an firstRow a companyName firstRow ? How to |
| IR[†] | Facebook C# SDK Get Current User |
| SUM-NN[†] (Rush et al., 2015) | how can i get the value of a string? |
| 2-layer BiLSTM | how to get the value of a cell in excel using c # |
| Transformer (Vaswani et al., 2017) | getting the first row in excel |
| TreeLSTM (Tai et al., 2015) | how to get the value of a cell in excel using c # |
| CodeNN[†] (Iyer et al., 2016) | how do I get the value of an xml file in c # ? |
| code2seq (this work) | get the value of a column in excel using c # |

```
static void Main(string[] args)
{
  // Create an instance of Bytescout.PDFRenderer.
  // RasterRenderer object and register it.
  RasterRenderer renderer = new RasterRenderer();
  renderer.RegistrationName = "demo";
  renderer.RegistrationKey = "demo";
  // Load PDF document.
  renderer.LoadDocumentFromFile("multipage.pdf");
  for (int i = 0; i < renderer.GetPageCount(); i++)
  {
    // Render first page of the document to BMP image file.
    renderer.RenderPageToFile(i, RasterOutputFormat.BMP,
      "image" + i + ".bmp");
  }

  // Open the first output file in default image viewer.
  System.Diagnostics.Process.Start("image0.bmp");
}
```

| Model | Prediction |
|---|---|
| MOSES[†] (Koehn et al., 2007) | How to add RasterRenderer renderer RasterRenderer renderer in a string in RegistrationName renderer Registration Key renderer LoadDocumentFromFile in C # How to a renderer Is a RenderPageToFile renderer in a string to BMP RasterOutputFormat each in C # |
| IR[†] | Select TOP 5 * from SomeTable, using Dataview.RowFilter? |
| SUM-NN[†] (Rush et al., 2015) | how do i create a text file in c# |
| 2-layer BiLSTM | how do i create a pdf file in c # ? |
| Transformer (Vaswani et al., 2017) | how to merge two pdf files ? |
| TreeLSTM (Tai et al., 2015) | how to make a process in c # |
| CodeNN[†] (Iyer et al., 2016) | How to get the value of an array in C # ? |
| code2seq (this work) | get the image from a pdf file in c # |

```
void Main() {
  string text = File.ReadAllText(@"T:\File1.txt");
  int num = 0;
  text = (Regex.Replace(text, "map", delegate(Match m) {
    return "map" + num++;
  }));
  File.WriteAllText(@"T:\File1.txt", text);
}
```

| Model | Prediction |
|---|---|
| MOSES[†] (Koehn et al., 2007) | How to File then How to HTML ? C # How to Write to |
| IR[†] | C# remove extra carriage returns from Stream |
| SUM-NN[†] (Rush et al., 2015) | how do i create a text file in c# |
| 2-layer BiLSTM | how to read a text file from a text file |
| Transformer (Vaswani et al., 2017) | how to write a . txt file in c # |
| TreeLSTM (Tai et al., 2015) | how to get the text from a text file in c # |
| CodeNN[†] (Iyer et al., 2016) | how to read a text file in c # ? |
| code2seq (this work) | replace a string in a text file |

Figure 6: C# examples from our test set for the code captioning task, along with the prediction of our model and each of the baselines.

```
void _______(Counter childCounter, String request, String requestId,
            long duration, boolean systemError, int responseSize) {
  // si je suis le counter fils du counter du contexte parent
  // comme sql pour http alors on ajoute la requête fille
  if (parentContext != null && parentCounter.getName()
    .equals(parentContext.getParentCounter().getChildCounterName())) {
      childHits++;
      childDurationsSum += (int) duration;
    }

  // pour drill-down on conserve pour chaque requête mère, les requêtes
  //  filles appelées et le nombre d'exécutions pour chacune
  if (parentContext == null) {
    addChildRequestForDrillDown(requestId);
  } else {
    parentContext.addChildRequestForDrillDown(requestId);
  }
}
```

| Model | Prediction |
|---|---|
| ConvAttention (Allamanis et al., 2016) | add |
| Paths+CRFs (Alon et al., 2018) | call |
| code2vec (Alon et al., 2019) | log response |
| 2-layer BiLSTM (no token splitting) | handle request |
| 2-layer BiLSTM | report child request |
| Transformer | add child |
| TreeLSTM (Tai et al., 2015) | add child |
| Gold: | add child request |
| code2seq (this work) | add child request |

```
public static int _______(int value) {
  return value <= 0 ? 1 :
    value >= 0x40000000 ? 0x40000000 :
      1 << (32 - Integer.numberOfLeadingZeros(value - 1));
}
```

| Model | Prediction |
|---|---|
| ConvAttention (Allamanis et al., 2016) | get |
| Paths+CRFs (Alon et al., 2018) | test bit inolz |
| code2vec (Alon et al., 2019) | multiply |
| 2-layer BiLSTM (no token splitting) | next power of two |
| 2-layer BiLSTM | { *(replaced UNK)* |
| Transformer | get bit length |
| TreeLSTM (Tai et al., 2015) | get |
| Gold: | find next positive power of two |
| code2seq (this work) | get power of two |

```java
BigInteger _______(int bitlength, BigInteger e, BigInteger sqrdBound)
{
  for (int i = 0; i != 5 * bitlength; i++)
  {
    BigInteger p = new BigInteger(bitlength, 1, param.getRandom());
    if (p.mod(e).equals(ONE))
    {
      continue;
    }
    if (p.multiply(p).compareTo(sqrdBound) < 0)
    {
      continue;
    }
    if (!isProbablePrime(p))
    {
      continue;
    }
    if (!e.gcd(p.subtract(ONE)).equals(ONE))
    {
      continue;
    }
    return p;
  }
  throw new IllegalStateException("unable to generate prime number..
    ...for RSA key");
}
```

| Model | Prediction |
|---|---|
| ConvAttention (Allamanis et al., 2016) | test |
| Paths+CRFs (Alon et al., 2018) | i |
| code2vec (Alon et al., 2019) | to big integer |
| 2-layer BiLSTM (no token splitting) | generate prime |
| 2-layer BiLSTM | generate prime number |
| Transformer | generate |
| TreeLSTM (Tai et al., 2015) | probable prime |
| Gold: | choose random prime |
| code2seq (this work) | generate prime number |

```java
public boolean _______(Set<String> set, String value) {
  for (String entry : set) {
    if (entry.equalsIgnoreCase(value)) {
      return true;
    }
  }
  return false;
}
```

| Model | Prediction |
|---|---|
| ConvAttention (Allamanis et al., 2016) | is |
| Paths+CRFs (Alon et al., 2018) | equals |
| code2vec (Alon et al., 2019) | contains ignore case |
| 2-layer BiLSTM (no token splitting) | contains ignore case |
| 2-layer BiLSTM | contains |
| Transformer | contains |
| TreeLSTM (Tai et al., 2015) | contains ignore case |
| Gold: | contains ignore case |
| code2seq (this work) | contains ignore case |

Figure 8: Java examples from our test set for the code summarization task, along with the prediction of our model and each of the baselines.

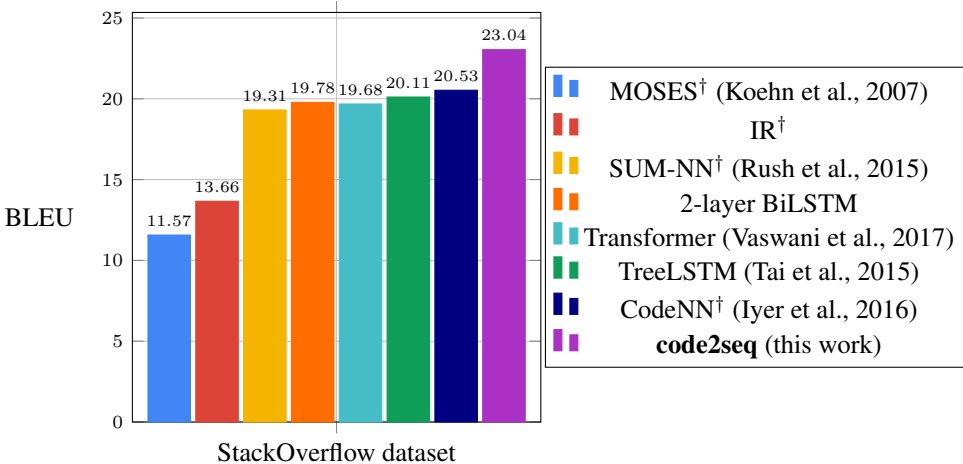

Figure 9: Visualization of the BLEU score of our model compared to the baselines, for the code captioning task. The values are the same as in Table 2. Our model achieves significantly higher results than the baselines.

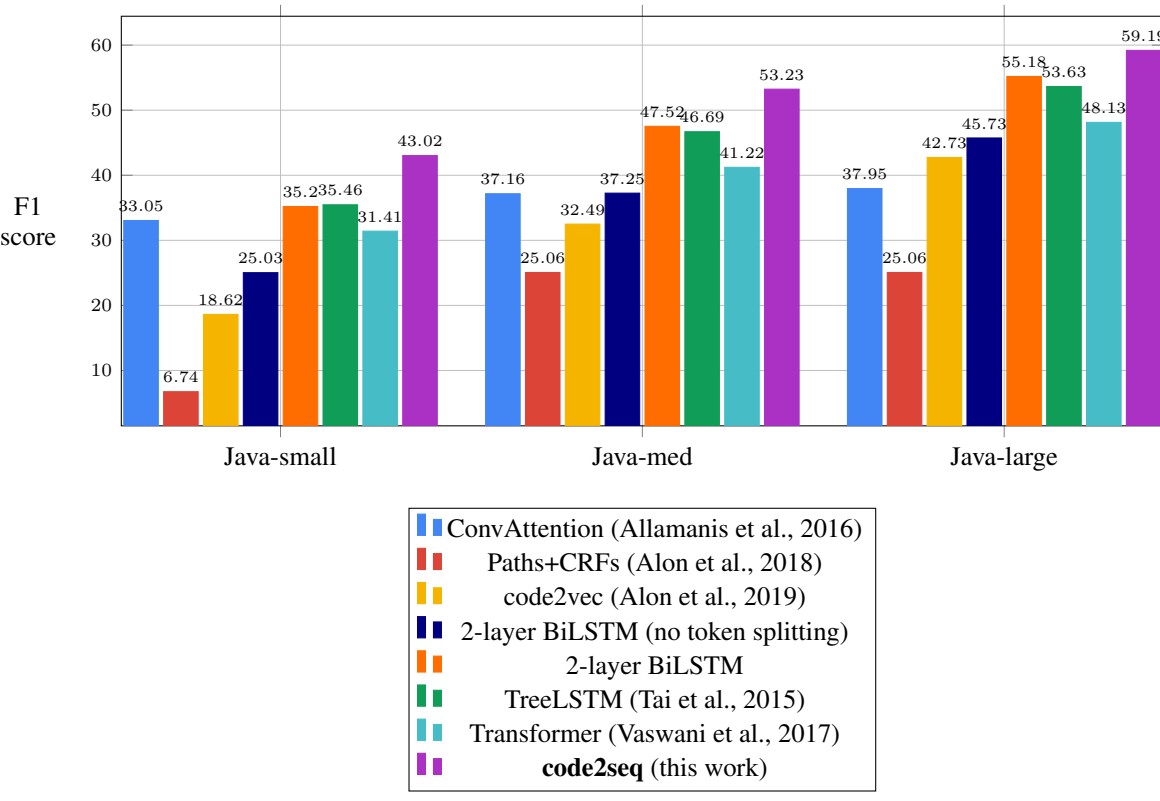

Figure 10: Visualization of the F1 score of our model compared to the baselines, for the code summarization task, across datasets. The values are the F1 columns from Table 1. Our model achieves significantly higher results than the baselines.

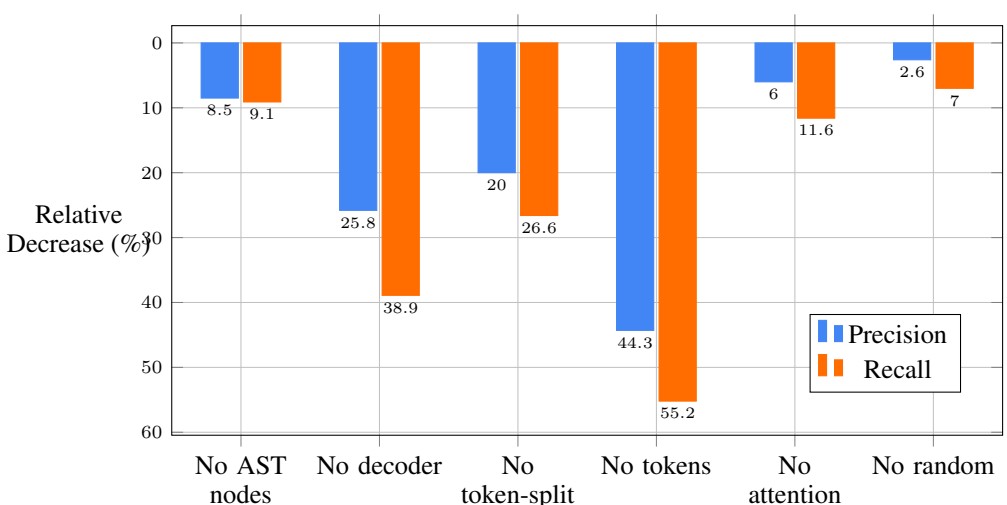

Figure 11: The relative decrease in precision and recall in each of the ablations, compared to the full model.

