# OpenReview forum: "code2seq: Generating Sequences from Structured Representations of Code"
_ICLR.cc/2019/Conference_

### Official Review · AnonReviewer3 · 2018-11-03
**Nice model using path summaries, but some baseline comparisons are missing**

**Rating:** 5
**Confidence:** 4

**Review:**

This paper presents a new code-to-sequence model called code2seq that leverages the syntactic structure of programming languages to encode source code snippets, which is then decoded to natural language using a sequence decoder. The key idea of the approach is to represent a program using a set of randomly sample k paths in its abstract syntax tree. For each path, the path is encoded using a recurrent network and concatenated with the embeddings of the two leaf terminal values of the path. The path encodings are then averaged to obtain the program embedding, which is then used to initialize a sequence decoder that also attends over the path embeddings. The code2vec model is evaluated over two tasks: 1) Code summarization: predicting a method’s name from its body, and 2) Code captioning: generating a natural language sentence from method’s body depicting its functionality. The code2seq model significantly outperforms the other baseline methods, and the ablation study shows the importance of various design choices.

This paper presents an elegant way to represent programs using a set of paths in the AST, which are then weighted using an attention mechanism to attend over relevant path components. The code2seq model is extensively evaluated over two domains of code summarization and code captioning, and results show significant improvements.

The novelty of the code2seq model is somewhat limited compared to the model presented in code2vec (Alon et al. 2018a) paper. In code2vec, a program is encoded as a set of paths, where each path comes from a fixed vocabulary. The code2seq model instead uses an LSTM to encode individual paths, which allows it to generalize to new paths. This is a more natural choice for embedding paths, but it doesn’t appear to be a big conceptual advance in the model architecture. The use of subtoken embeddings for encoding/decoding identifier names is different in code2seq, but it has been proposed earlier in other code embedding models.

For the code summarization evaluation, would it be possible to evaluate the code2seq model on the dataset used by the code2vec paper? On that dataset, the code2vec approach gets a precision score of 63.1, recall of 54.4, and F1 score of 58.4, [Table 3 on page 18] which are comparable to overall scores of the code2seq model.

One of the key findings of the paper is that syntactic structure of programs is important to encode. Similar observations have been made in other program embedding papers that use for example Tree-RNN [1] or graph neural networks (GNN) [Allamanis et al. 2018]. It would be quite valuable to compare the current results with the Tree-RNN or GNN models (without performing additional dataflow and control-flow post processing) to see how well the paths-based embeddings work in comparison to these models.

The value of k=200 seems a bit large for the examples presented in the paper. What happens when smaller values of k are used (e.g. k=10, 20?) What are the average number of paths in the java programs in the dataset?

1. Chris Piech, Jonathan Huang, Andy Nguyen, Mike Phulsuksombati, Mehran Sahami, Leonidas Guibas. Learning Program Embeddings to Propagate Feedback on Student Code
ICML 2015

---

> ### Author Response · Authors · 2018-11-08
> **Authors response**
>
> Thank you for your detailed review. You raise many important points, which we think are all addressable within the discussion phase. Please see our detailed response below.
>
> | The novelty of the code2seq model is somewhat limited compared to the model presented in
> | code2vec (Alon et al. 2018a) paper.
>
> The main difference between code2vec and code2seq is inherent even in the name: our model generates a sequence, whereas code2vec is a classifier. While the two share some ideas on the encoder side, to say that our code2seq approach is not novel because code2vec already exists is akin to saying that seq2seq NMT is not novel because we already had text categorization models.
> All of these advantages over code2vec give our model 40%-230% better results than code2vec on the code summarization task, while code2vec definitely cannot perform the code captioning task (Section 4.2). Our work sets a new state of the art for code-to-natural-language tasks, with a model that is much more compact in terms of the number of parameters (code2seq has an order-of-magnitude fewer parameters compared to code2vec!)..
>
> | For the code summarization evaluation, would it be possible to evaluate the code2seq model
> | on the dataset used by the code2vec paper? On that dataset, the code2vec approach gets a
> | precision score of 63.1, recall of 54.4, and F1 score of 58.4.
>
> We agree that this is a little confusing, we will clarify the following point in the paper:
> on their dataset, our model gets precision of 70.2 (vs. their 63.1), recall of 63.3 (vs. their 54.4) and F1 of 66.6 (vs. 58.4). We believe that the reason for the lower results of code2vec on our datasets is that their dataset is split to train/dev/test *by-file*, while in our datasets we split always *by project*. In their dataset, a file can be in the training set, while another file from the same project can be in the test set. This makes their dataset significantly easier, because method names often "leak" to other files in the same project, and there are often duplicates in different files of the same project. This is consistent with Allamanis et al (ICLR'2018) who found that splitting by-file makes the dataset easier than by-project.
> We decided to take the stricter approach, and not to use their dataset (even though our model achieves better results on it), in order to make all of our comparisons on split-by-project datasets. We will add the results on their per-file split dataset to the appendix, although we advocate for using the harder per-project split.
>
>
> | One of the key findings of the paper is that syntactic structure of programs is important to
> | encode. Similar observations have been made in other program embedding papers that use
> | for example Tree-RNN [1] or graph neural networks (GNN) [Allamanis et al. 2018]. It would be
> | quite valuable to compare the current results with the Tree-RNN or GNN models
>
> We did our best to find and re-train *any* baselines that we could find and are relevant for the task of generating sequences from code. We could not compare directly to [1] (ICML'2015) or the GGNN work (Allamanis et al 2018) because they cannot generate sequences and are thus incomparable to our work.
> We think that deliberately paralyzing our model's decoder and attention mechanism just to compare to older models (from 2015) and tasks goes against the main idea of our work: to generate sequences from code - a task which none of these works solve.
>
>
> | The value of k=200 seems a bit large for the examples presented in the paper. What happens
> | when smaller values of k are used (e.g. k=10, 20?) What are the average number of paths in
> | the java programs in the dataset?
>
> If an example contains less than 200 paths, we simply take all of them. A "too large" value of k is not a problem.
> The average number of paths in our Java-large training set is 220 paths per example. For some large methods, a number as high as 200 is beneficial. We empirically experimented with the value of k but did not include it because of space limitations. Lower values than k=100 show worse results, increasing to k=200 shows a minor improvement, above k=200 there is no improvement. Practically, k=200 was found to be a reasonable sweet spot between capturing enough information while keeping training feasible in the GPU's memory.
> We will include those and other statistics of the datasets.

---

> > ### Comment · AnonReviewer3 · 2018-11-18
> > **Thanks for the response; some additional clarification**
> >
> > Thanks for the clarification regarding the results presented in code2vec (Alon et al. 2018a) paper, and the choice of different k values. I think it might be interesting to add results (with accuracy numbers) regarding different k values.
> >
> > | The main difference between code2vec and code2seq is inherent even in the name: our model generates a
> > | sequence, whereas code2vec is a classifier. While the two share some ideas on the encoder side, to say that our
> > | code2seq approach is not novel because code2vec already exists is akin to saying that seq2seq NMT is not novel
> > | because we already had text categorization models.
> >
> > Yes, the difference between generating a single output and sequence is important, but isn't the decoder simply an LSTM generating sequence of sub-tokens while attending over the path summaries? Since a similar decoder was proposed in  Allamanis et al. (2016) to generate sequences of sub-tokens, I was focusing the novelty more on the encoder side of the model (which is quite interesting), but there it seems like a small modification over the code2vec paper.
> >
> > | We could not compare directly to [1] (ICML'2015) or the GGNN work (Allamanis et al 2018) because they cannot
> > | generate sequences and are thus incomparable to our work. We think that deliberately paralyzing our model's
> > | decoder and attention mechanism just to compare to older models (from 2015) and tasks goes against the main
> > | idea of our work: to generate sequences from code - a task which none of these works solve.
> >
> > I am not sure I fully understand this comment. Why can't the encoder be a Tree-RNN [1] (ICML'2015) or a GGNN (Allamanis et al 2018) with a similar LSTM decoder over sub-tokens? I believe one of the key findings of the paper is that the syntactic structure of programs is important, and the present encoder does a great job at representing programs using a summary of randomly selected paths. But these other papers also use different models to capture the structure of programs and show that it helps there as well. Without comparisons to them, it is difficult to know whether those models are better at capturing syntactic structure of programs or if the code2seq encoder model is better.

---

> > > ### Author Response · Authors · 2018-11-20
> > > **Authors' response**
> > >
> > > Thank you for your questions and comments.
> > >
> > > | novelty on the encoder side ... there it seems like a small modification over the code2vec
> > > | paper.
> > > Besides the decoder, our model has several conceptual advantages over code2vec: it is an order of magnitude lighter, and it can represent *any* path while code2vec can represent only monolithically observed paths. Additionally, it performs better than code2vec even without a decoder. Although it shares some ideas with code2vec, our work is a milestone which further increases our understanding of the abilities of neural networks in this area of encoding source code and transforming it to other modalities.
> > >
> > > | Why can't the encoder be a Tree-RNN (ICML'2015) or a GGNN (Allamanis et al 2018) with a
> > > | similar LSTM decoder over sub-tokens?  ... Without comparisons to them, it is difficult to know
> > > | whether those models are better at capturing syntactic structure of programs or if the code2seq
> > > | encoder model is better.
> > > The GGNN model of Allamanis et al (ICLR'2018) cannot encode an entire AST as our model. In that model, every identifier in the program is assigned a graph node. For example, in the statement "x = y;" each of "x" and "y" is assigned a graph node, with an edge that represents "ComputedFrom" between them. Then, representations of these identifiers are updated by propagating information from their neighbor nodes. The final outcome is a vector representation for each node (i.e., a variable or a constant), but there is no representation for the entire program. Thus, it is not a substitution for a program encoder, and adapting it to encode a whole AST is an interesting, non-trivial, direction on its own.
> > >
> > > We agree that Tree-RNNs can encode source code and use an LSTM decoder, and we will include a reference to the ICML'2015 work. However, the fact remains that no work has done that for the difficult task of generating sequences. The closest work which encoded code using Tree-RNNs and generated sequences is the work of Liang et al. (AAAI'2018) - which did not show promising results: their model performed better than NMT baselines only in some of their benchmarks and only because they hindered their baselines by depriving them of non-alphanumerical characters. We agree that exploring other approaches for encoding source code is a fascinating direction for future research.

---

### Official Review · AnonReviewer1 · 2018-11-03
**Nice evaluation of AST-based encoding for code summarization tasks**

**Rating:** 7
**Confidence:** 4

**Review:**

This paper introduces an AST-based encoding for programming code and
shows the effectivness of the encoding in two different task of code
summarization:

1. Extreme code summarization - predicting (generating) function name from function body (Java)
2. Code captioning - generating a natural language sentence for a (short) snippet of code (C#)

Pros:
- Simple idea of encoding syntactic structure of the program through random paths in ASTs
- Thorough evaluation of the technique on multiple datasets and using multiple baselines
- Better results than previously published baselines
- Two new datasets (based on Java code present in github) that will be made available
- The encoding is used in two different tasks which also involve two different languages

Cons:
- Some of the details of the implementation/design are not clear (see some clarifying questions below)
- More stats on the collected datasets would have been nice
- Personally, I'm not convinced "extreme code summarization"
is a great task for code understanding (see more comments below)

Overall, I enjoyed reading this paper and I think the authors did a
great job explaining the technique, comparing it with other baselines,
building new datasets, etc.

I have several clarifying questions/points (in no particular order):

* Can you provide some intuition on why random paths in the AST encode
  the "meaning" of the code? And perhaps qualitatively compare it with
  recording some other properties from the tree that preserve its
  structure more?

* When you perform the encoding of the function body, one sample in a
  training step contains all the k (k = 200) paths and all the 2*k
  terminals (end of Section 2)? Or one path at a time (Section 3.2)?
  I'm guessing is the latter, but not entirely sure. Figure 3 could
  improve to make it clear.

* Can you explain how you came up with k = 200? I think providing some
  stats on the dataset could be helpful to understand this number.

* The results for the baselines - do you train across all projects?
  (As you point out, ConvAttention trained separately, curious whether
  it makes a difference for the 2 datasets med and large not present
  in the original paper).

* I'm not sure I understand parts of the ablation study. In particular
  for point 1., it seems that instead of the AST, only the terminal
  nodes are used. Do you still use 200 random pairs of terminal? Is
  this equivalent to a (randomly shuffled) subset of the tokens in the
  program? Also could you explain why you do the ablation study on the
  validation set of the medium dataset? In fact, the caption of Table
  3 says it's done on the dev set. This part was a bit confusing.

* I would have liked to see more details on the datasets introduced,
  in particular wrt metrics that are relevant for training the model
  you describe (e.g., stats on the ASTs, stats on the number of random
  paths in ASTs, code length in tokens, etc.)

* I'm not convinced that the task of "extreme code summarization" is a
  meaningful task. My main problem with it is that the performance of
  a human on this task would not be that great. On one hand humans
  (I'm referring to "programming humans" :) ) have no problem in
  coming up with a name for a function body; however, I'm not
  convinced they could predict the "gold" standard. Or, another way of
  thinking about this, if you have 3 humans who provided names for the
  same function, my guess it that there will be a wide degree of
  (dis)agreement. Some of the examples provided in the supplementary
  material can serve as confirmation bias to my thought :): Fig 7. I
  claim "choose random prime" and "generate prime number" are
  semantically close, however, the precision and recall for this
  example are both low. All this being said, I understand that it's a
  task for which data can be generated fairly quickly to feed the
  (beast) NN and that helps pushing the needle in understanding code
  semantics.

* It would be nice to see "exact match" as one of the metrics (it is
  probably low, judging by F1 scores, but good to be reported).

* Most likely the following paper could be cited in the related work:
Neural Code Comprehension: A Learnable Representation of Code Semantics
https://arxiv.org/abs/1806.07336
https://nips.cc/Conferences/2018/Schedule?showEvent=11359

Page 5 first phrase at the top, perhaps zi is a typo and it is
supposed to be z1?

----

Update: after all the discussion, I'm lowering my score a bit while still hoping the paper will get published. I'm satisfied with the results and the improvement of the paper. I still find it a bit surprising that the pairs of literals/leaves in the tree are a good approximation for the program itself (as shown in one of the ablation study).

---

> ### Author Response · Authors · 2018-11-08
> **Authors Response 1/2**
>
> Thank you for reviewing our work so kindly.
>
> | Can you provide some intuition on why random paths in the AST encode the "meaning" of the code?
>
> We think that AST paths are a useful decomposition of the AST because they capture long-range relations and interactions in a relatively short sequence of symbols. For example, the green path (3) in Figure 1(a) tells us that a variable named "set" is iterated in a "For" loop, and then the method "equalsIgnoreCase" is called on an object inside the condition of an "If" statement, which resides inside the loop body. This path contains a lot of information in a relatively short sequence of symbols (6 nodes + 2 terminal values). Additionally, these paths are robust to mutations - even if we inserted an additional statement between the "For" and the "If" lines - this path will still be kept (in contrast to the sequential NMT baselines, in which insertions will push "set" and "equalsIgnoreCase" farther away from each other).
>
> | When you perform the encoding of the function body, one sample in a training step contains all
> | the k (k = 200) paths and all the 2*k terminals (end of Section 2)? Or one path at a time (Section 3.2)?
>
> One sample in a training step contains all the k=200 sampled paths and 2*k terminals. In Figure 3, the green boxes are all the k=200 paths with their terminals.
> The attention mechanism is used to dynamically select a distribution over these 200 paths while decoding, just like a NMT model would attend over the source tokens.
> We will clarify this in Section 3.2.
>
>
> | Can you explain how you came up with k = 200? I think providing some stats on the dataset could be helpful to understand this number.
>
> The average number of paths in our Java-large training set is 220 paths per example. For some large methods, a number as high as 200 is beneficial. We empirically experimented with the value of k but did not include it because of space limitations. Lower values than k=100 show worse results, increasing to k=200 shows a minor improvement, and for k>200 there is no significant improvement. Practically, k=200 was found to be a reasonable sweet spot between capturing enough information while keeping training feasible in the GPU's memory.
> We will include those and other statistics of the datasets.
>
> | The results for the baselines - do you train across all projects? (As you point out,
> | ConvAttention trained separately, curious whether it makes a difference for the 2 datasets med
> | and large not present in the original paper).
>
> In all of our experiments we always train across all projects. ConvAttention trained separately in their original paper, but when we retrained their model we trained it (and our model) across multiple projects and tested on other projects. Training and testing within the same project makes the problem significantly easier (as Allamanis et al (ICML'2016) also note). In our benchmarks, we test how well a model can generalize to completely unseen projects.
>
> | I'm not sure I understand parts of the ablation study. In particular for point 1., it seems that
> | instead of the AST, only the terminal nodes are used. Do you still use 200 random pairs of
> | terminal? Is this equivalent to a (randomly shuffled) subset of the tokens in the program?
>
> We still use 200 random pairs of terminals, yes. Ablation #1 (no AST nodes) is equivalent to a random subset of *pairs* of tokens, which is more informative than just a subset of the tokens. We hypothesize that the fully connected layer learns how each pair of tokens interact with each other, and attending over those interactions it is a little more powerful than attending over each token separately.
>
> | why you do the ablation study on the validation set of the medium dataset? In fact, the caption
> | of Table 3 says it's done on the dev set.
>
> The choice for the dataset in the ablation study could have been different - we hoped that since Java-med contains the top 1000 projects in Github, it is both large enough and of a high enough code quality to easily observe the effect of the ablations.
> We performed it on the validation set of Java-med. The development (dev) set refers to the validation set; we will clarify the terminology in Table 3.
>
>
> | I would have liked to see more details on the datasets introduced, in particular wrt metrics that
> | are relevant for training the model you describe (e.g., stats on the ASTs, stats on the number of
> | random paths in ASTs, code length in tokens, etc.)
>
> We agree. We will include additional stats.

---

> > ### Author Response · Authors · 2018-11-08
> > **Authors response 2/2**
> >
> > |  I'm not convinced that the task of "extreme code summarization" is a meaningful task. My
> > | main problem with it is that the performance of a human on this task would not be that great.
> > | On one hand humans (I'm referring to "programming humans" :) ) have no problem in coming
> > | up with a name for a function body; however, I'm not convinced they could predict the "gold"
> > | standard. Or, another way of thinking about this, if you have 3 humans who provided names for
> > | the... All this being said, I understand that it's a task for which data can be generated fairly
> > | quickly to feed the (beast) NN and that helps pushing the needle in understanding code semantics.
> >
> > We agree that code summarization tasks are difficult to measure and evaluate. For this reason, we follow the standard practice of the existing literature and used tasks and metrics that were introduced by previous work (Iyer et al., ACL 2016; Allamanis et al. ICML 2016). Since the improvement of our model over the baselines is substantial and consistent across datasets and tasks, we believe that our model is better at modeling the data and is thus an important contribution.
> >
> > | It would be nice to see "exact match" as one of the metrics (it is probably low, judging by F1 scores, but good to be reported).
> >
> > Exact match accuracy for our model: on Java-large: 35.0%, Java-med: 29.5%, Java-small: 15.4%. We did not include these results because they are mostly correlated with the F1 scores, and we feel that precision/recall is more informative.
> >
> > |  Most likely the following paper could be cited in the related work: Neural Code Comprehension: A Learnable Representation of Code Semantics
> >
> > Thank you, we will include a reference.
> >
> > | Page 5 first phrase at the top, perhaps zi is a typo and it is supposed to be z1?
> >
> > Thank you, this is indeed a typo and should be z1.

---

### Official Review · AnonReviewer2 · 2018-11-06

**Rating:** 6
**Confidence:** 4

**Review:**

The authors present a method for generating sequences from code. To achieve this, they parse the code and produce a syntax tree. Then, they enumerate paths in the tree along leaf nodes. Each path is encoded via an bidirectional LSTM and a (sub)token-level LSTM decoder with attention over the paths is used to produce the output sequence.  The authors compare their model with other models and show that it outperforms them on two code-to-sequence tasks. An ablation study shows how different components affect the model's performance.

Overall, the task seems very interesting and the results positive. My main concern is wrt the novelty of this work: the novelty of the proposed model seems limited compared to code2vec (Alon 2018b). To my understanding the core idea of both code2vec and code2seq is similar in many respects. The core difference is that paths, instead of treated as single units (code2vec), they are treated as sequences whose representation is computed by an LSTM.

To understand the work better, additional evaluation seem be necessary:

Q1: Could the authors compare code2seq with an ablation of a 2-layer BiLSTM where the decoder predicts the output as a single token (similar to the "no decoder" ablation of code2vec)?

Comparing this result to the "no decoder" ablation of code2seq will show the extent to which code2seq's performance is due to its code encoding or if code2vec with an LSTM decoder output would have sufficed.

Q2: Using the BiLSTM and the Transformer as baselines seems reasonable but there are other existing models such as Tree LSTMs, Graph Convolutional Neural Networks [a] and TBCNNs [b] that could also be strong baselines which take tree structure into account. Have the authors experimented with any of those?

Q3: I find the results in Table 1 very confusing when comparing them with those reported in Alon et al(2018b): code2vec achieves the best performance in Alon et al (2018b) but it seems to be performing badly in this work. The empirical comparisons to the same baseline methods used in Alon et al. (2018b) yield very different results. Why is that so? It would be worth performing an additional evaluation on the datasets of Alon et al (2018b) using the code2seq model. This would clarify if the results observed here generalize to other datasets.

Q4: The strategy of enumerating paths in the tree seems to be problematic for large files of code. It is unclear how the authors (a) do an unbiased sample of the paths. Do they need to first enumerate all of them and pick at uniform? (b) since the authors pick $k$ paths for each sample, this may imply that the larger the tree, the worse the performance of code2seq. It would be useful to understand if code2seq suffers from this problem more/less than other baselines.

[a] Kipf, T.N. and Welling, M., 2016. Semi-supervised classification with graph convolutional networks.
[b] Mou, L., Men, R., Li, G., Xu, Y., Zhang, L., Yan, R. and Jin, Z., 2015. Natural language inference by tree-based convolution and heuristic matching.

---

> ### Author Response · Authors · 2018-11-08
> **Authors response**
>
> Thank you for your detailed review. You raise many important points, which we think are all addressable within the discussion phase. Please see our detailed response below.
>
> | Q1: Could the authors compare code2seq with an ablation of a 2-layer BiLSTM where the
> | decoder predicts the output as a single token (similar to the "no decoder" ablation of
> | code2vec)?
> | Comparing this result to the "no decoder" ablation of code2seq will show the extent to which
> | code2seq's performance is due to its code encoding or if code2vec with an LSTM decoder
> | output would have sufficed.
>
> We definitely can. The results for this baseline (2-layer BiLSTM encoder with single-token prediction, with the same size of target vocab as in our "no decoder") are:
> Precision: 31.42, Recall: 13.92, F1: 19.29
> Our model with "no decoder" on the same dataset (Table 3):
> Precision: 47.99, Recall: 28.96, F1: 36.12.
>
> | Q2: Using the BiLSTM and the Transformer as baselines seems reasonable but there are
> | other existing models such as Tree LSTMs, Graph Convolutional Neural Networks [a] and
> | TBCNNs [b] that could also be strong baselines which take tree structure into account.
>
> We did our best to find and re-train *any* baselines that we could find and are relevant for the task of generating sequences from code. We could not compare to the Tree-RNN of Liang and Zhu (AAAI'2018) because of replicability issues in their work (as described in the footnote of page 5); the Graph Convolutional model of [a] was designed for classification only; the TBCNN work of [b] addressed sentiment *classification* in NLP; the model of Mou et al (AAAI'2016) is also a classifier and was applied to code classification tasks and to detection of bubble sort. All of these classification models cannot generate sequences and are thus incomparable to our work.
> We think that deliberately paralyzing our model's decoder and attention mechanism just to compare to older models (from 2015-2016) and tasks goes against the main idea of our work: to generate sequences from code - a task which none of these works solve.
>
> | Q3: code2vec achieves the best performance in Alon et al (2018b) but it seems
> | to be performing badly in this work.
>
> We agree that this is a little confusing, we will clarify the following point in the paper:
> on their dataset, our model gets precision of 70.2 (vs. their 63.1), recall of 63.3 (vs. their 54.4) and F1 of 66.6 (vs. 58.4). We believe that the reason for the lower results of code2vec on our datasets is that their dataset is split to train/dev/test *by-file*, while in our datasets we split always *by project*. In their dataset, a file can be in the training set, while another file from the same project can be in the test set. This makes their dataset significantly easier, because method names often "leak" to other files in the same project, and there are often duplicates in different files of the same project. This is consistent with Allamanis et al (ICLR'2018) who found that splitting by-file makes the dataset easier than by-project.
> We decided to take the stricter approach, and not to use their dataset (even though our model achieves better results on it), in order to make all of our comparisons on split-by-project datasets. We will add the results on their per-file split dataset to the appendix, although we advocate for using the harder per-project split.
>
> | Q4: The strategy of enumerating paths in the tree seems to be problematic for large files of
> | code. It is unclear how the authors (a) do an unbiased sample of the paths. Do they need to
> | first enumerate all of them and pick at uniform? (b) since the authors pick $k$ paths for each
> | sample, this may imply that the larger the tree, the worse the performance of code2seq.
>
> Large files are not a problem because we worked on method level, but we agree that maybe extremely huge *methods* are.
> (a) We enumerate all of the paths in advance and sample them uniformly at training time (a different subset is sampled on every training iteration). This is a technical detail; in a future implementation we plan to generate random paths on CPU in parallel to training on GPU.
> (b) We did observe that scores on larger Java methods are a little lower than on short methods, but mostly because the names are more diverse in long methods. This was even more apparent for the seq2seq baselines, because large Java methods are sometimes more than 5000 tokens long, and are thus very difficult to digest for the BiLSTM and the Transformer.
> Since we evaluated our model and the baselines on StackOverflow snippets and Java methods, the size of the code was not much of a problem. If a future dataset contained extremely large code snippets, the value of k can be easily increased (maybe at the cost of a smaller batch size). Remember that k does not need to be as large as the number of existing paths, since the information contained in a missing path is often "covered" by other paths.

---

> > ### Comment · AnonReviewer2 · 2018-11-14
> > **Reviewer Response**
> >
> > Thank you for the response. You have addressed many of my questions and I have increased my rating by one. However, I would like to further discuss Q2 and Q4, which I find important.
> >
> > Q2: Indeed the cited papers are not directly applicable to your task and would require some small modifications, which nevertheless are not an unreasonable ask for an ICLR submission. Also, there are papers based on the principles of [a] and [b] that are directly applicable. This is a point that AnonReviwer3 also discusses.
> >
> > * Chen et al. [c] can be used as a TreeLSTM-based baseline for your task. While their model's decoder is a TreeLSTM one can easily pass a sequence as a degenerate tree to the decoder, without even modifying the code. Removing the existing decoder for an LSTM should not be hard either.
> > * Bastings et al. [d] has a GCN-based encoder/LSTM-decoder baseline that seems directly applicable to your setting.
> >
> > Comparing with these models will allow readers to understand the trade-offs involved and the weaknesses of each model.
> >
> > Q4: Your response makes sense and although I won't insist on this, it would be nice to backup your claim with a graph that shows how the performance of the different models decreases as the size of the input method grows.
> >
> >
> >
> > [c] Chen, X., Liu, C. and Song, D., 2018. Tree-to-tree Neural Networks for Program Translation. arXiv preprint arXiv:1802.03691.
> > [d] Bastings, J., Titov, I., Aziz, W., Marcheggiani, D. and Sima'an, K., 2017. Graph convolutional encoders for syntax-aware neural machine translation. arXiv preprint arXiv:1704.04675.

---

> > > ### Author Response · Authors · 2018-11-18
> > > **Authors' Response**
> > >
> > > Thank you for increasing your rating and for your suggestions.
> > >
> > > Q2: Thank you for both references. We contacted Chen et al, but they were not able to provide their code at this time.
> > > As for replicating additional NMT baselines: there are dozens of interesting NMT approaches presented at each conference, and it is not reasonable for us to replicate them all for our code2seq benchmarks. Therefore, we focused on the two most popular seq2seq approaches, i.e., LSTMs and transformers.
> > >
> > > It is also important to emphasize that applying graph convolutions to code is not a straightforward application of Bastings et al's approach. For example, a fundamental difference between dependency trees in natural language and ASTs is that all non-terminals in an AST are of a very limited vocabulary, whereas almost any natural-language word can be a non-terminal in a dependency parse. We agree that exploring graph convolutions for the task of code-to-sequence is an interesting direction for future research.
> > >
> > > Q4: This is a great idea that will further strengthen our paper. We will add a graph that shows how the performance of different models changes as the size of the input method grows. This will be added within this discussion period.

---

> > > > ### Comment · AnonReviewer2 · 2018-11-19
> > > > **Thanks**
> > > >
> > > > Thank you for addressing my comments. Regarding Q2, I understand your position and the unfortunate situation regarding the code of Chen et al.
> > > >
> > > > Although I still think that a comparison with TreeLSTMs and/or GCNs would be useful, taking into consideration the unavailability of Chen et al. code, if the AC and the other reviewers do not see this as an issue and agree with the authors' position that the comparison with existing baselines is sufficient, I will _not_ further argue for rejecting this paper.

---

> > > > > ### Author Response · Authors · 2018-11-20
> > > > > **Thank you**
> > > > >
> > > > > Thank you.
> > > > > We included references to Chen et al. and Bastings et al. in the Related Work section.

---

### Comment · Area_Chair1 · 2018-11-09
**Two quick clarifications**

Thank you for the quick and thorough author response, I trust it has addressed some reviewer concerns. (Reviewers, please take not and respond/change scores as necessary!) I just had two other quick clarifications.

First, the paper says "To the best of our knowledge, this is the first work to leverage the syntactic structure of code for end-to-end generation of sequences." I'm not sure if this is 100% true, and it should probably be clarified with respect to the following two works:
* One of the first works on using machine learning methods for code commenting, albeit not in the neural framework, used a model that transformed AST trees into pseudocode: "Learning to Generate Pseudo-code from Source Code using Statistical Machine Translation," Oda et al. ASE 2015.
* Recently, there has been a paper that used AST structure, although linearized into a sequence, for neural code comment generation: "Deep Code Comment Generation," Hu et al. ICPC 2018.
The second is an easy baseline that could be added, and the first has been tested on the publicly available Django code commenting dataset, so an empirical comparison may be feasible.

Second, I had a question about the ablation:
I found it quite puzzling that the "No AST Nodes (only tokens)" baseline was highly competitive, better than any of the other tested methods. Do you have any idea why this would be the case? I couldn't think of any reason why a method that only looked at pairs of tokens would do better than a method like the transformer, which has much more expressive power, and can implicitly capture pairs of tokens through self attention.

---

> ### Author Response · Authors · 2018-11-11
> **Response**
>
> Thank you for reading our work and responses, and for your insightful comments.
>
> | The paper says "To the best of our knowledge, this is the first work to leverage the syntactic
> | structure of code for end-to-end generation of sequences." I'm not sure if this is 100% true.
>
> We agree. We will refine our claim to "the first work to directly use paths in the abstract syntax tree for end-to-end generation of sequences". We will cite those works, thank you for bringing them to our attention.
>
> | "Learning to Generate Pseudo-code from Source Code using Statistical Machine Translation,"
> | Oda et al. ASE 2015.
>
> Thank you for bringing this work to our attention. This may have been the first to generate sequences by leveraging the syntax. The problem in comparing our work with Oda et al.'s is that they perform line-by-line statistical machine translation (SMT), and can thus leverage the given line-alignment between the source code and target pseudocode. Our tasks are different, and we cannot assume a 1-to-1 alignment between elements in the input and the output; our tasks take a whole code snippet as their input and produce a much shorter sequence as output.
> A conceptual advantage of our model over line-by-line translation is that our model can capture multiline patterns in the source code. These multiline patterns are often very useful for the model and get the most attention (Figure 1(a)).
> Practically, we will not manage to adapt their model to Java/C# or adapt our model to Python within the discussion period. The closest baseline that we have is the MOSES SMT tool, which our model outperforms by a large gap. We will cite this work and discuss the differences.
>
> | "Deep Code Comment Generation," Hu et al, ICPC'2018
>
> This would be an interesting comparison; however, neither their code nor dataset are publicly available. We just emailed the author and asked for the code or dataset, and we will add this as a baseline if possible within the discussion period.
> As you suggest, there is a conceptual difference between our approaches. Hu et al. linearize the AST, and then pass it on to a standard seq2seq model. We present a new model, in which the encoder already assumes that the input is tree-structured. We will cite this work and discuss the differences.
>
> |  the "No AST Nodes (only tokens)" baseline was highly competitive, better than any of the
> | other tested methods. Do you have any idea why this would be the case? I couldn't think of
> | any reason why a method that only looked at pairs of tokens would do better than a method
> | like the transformer, which has much more expressive power, and can implicitly capture pairs
> | of tokens through self attention.
>
> This surprised us as well. The transformer is indeed able to capture all pairs of tokens. However, *not all tokens are AST leaves*. By focusing on AST leaves, we are increasing the focus on named tokens, and effectively ignoring functional tokens like brackets, parentheses, semicolons, etc.
> Transformers can (in theory) capture the same signal, but perhaps they require significantly more layers or a different optimization to actually learn to focus on those particular elements. The AST gives us this information for free without having to spend more transformer layers just to learn it.
> Additionally, for practical reasons, we limited the length of the paths to 9. This leads to pairs of leaves that are close in the AST, but not necessarily close in the sequence. In contrast, the transformer's attention is effectively skewed towards sequential proximity because of the positional embeddings.
> We will include this discussion in our ablation study.

---

### Author Response · Authors · 2018-11-11
**Submission update 11/11/2018 - Summary of changes**

We updated our submission to address some of the comments raised by the reviewers and the AC:

- We improved our wording in "the first work to leverage the syntactic structure of code for end-to-end generation of sequences" (Introduction), as suggested by the AC.
- We included references to Oda et al. (ASE'2015), Hu et al. (ICPC'2018) and discussed the differences between our work and theirs in the Related Work [AC].
- We further discussed the "no AST nodes" ablation and compared it conceptually to the Transformer in the Ablation Study [Reviewer1, AC].
- We clarified that we use k paths for each sample in a single training step (and not only a single path) in Section 3.2 and Figure 3 [Reviewer1].
- We discussed the choice of k=200 in Appendix A, with a reference from Section 4 (this was included in the initial version in Appendix E but was maybe easy to miss.) [Reviewer1,3]
- We clarified that we train across all projects, rather than separately. [Reviewer1]
- We clarified that the ablation study was performed on the *validation set* of Java-med in Table 3 (instead of "dev set"). [Reviewer1]
- We added a table of statistics of all of the datasets in Appendix A with a reference from the Evaluation section. [Reviewer1,3]
- We added an empirical comparison to code2vec *on their dataset* and a discussion of splitting by-project compared to by-file in Appendix A. [Reviewer2+3].
- Fixed a typo in page 5 (k_1 instead of k_i) [Reviewer1]

Thank you again for your fruitful comments.

---

### Author Response · Authors · 2018-11-18
**Submission update 11/18/2018 - Summary of changes**

We updated our submission to include a great suggestion by Reviewer 2:

- We added a graph of performance compared to the number of input code lines to Appendix A (page 13), with a reference from Section 4.
- Table of statistics (Table 5): we added the average number of lines in each dataset (tl;dr: around 6.5 in Java methods, and 8.3 in C# StackOverflow posts )

Thank you for this fruitful suggestion.

---

### Author Response · Authors · 2018-11-20
**Summary of additional experiments**

We would like to thank the reviewers for all their excellent suggestions! We feel that they have helped us improve the paper.

Following comments and suggestions by the reviewers, we performed the following major experiments so far:
* Performed an evaluation of our model on code2vec's dataset (which showed an improvement over their results in 8.2 F1 points).
* Performed an evaluation of our model on Hu et al.'s (ICPC'2018) dataset (which showed an absolute gain of 5.5 BLEU points, 60% relative).
* Performed an additional experiment of BiLSTMs without a decoder (which showed that our model without a decoder achieves an absolute gain of 16 F1 points over a BiLSTM without a decoder).
* Performed an additional analysis of our test set with respect to input code size (which showed that our model consistently outperforms the baselines across short and long code snippets).

* We could not directly compare to the following suggested works:
   + GGNNs (Allamanis et al., ICLR'2018) - because their model is not able to generate sequences nor to encode a whole snippet of code, but can only represent atomic elements such as variables.
   + Tree2tree (Chen et al., 2018) is not publicly available and addresses a different task.
   + Related works such as Piech et al. (ICML'2015), Kipf et al. (ICLR'2017), Mou et al. (AAAI'2016) are all interesting directions but do not generate sequences, and adapting them to our tasks is a contribution of its own. Bastings et al.'s (EMNLP'2017) approach cannot be applied to source code in a straightforward way.

We included references to Chen et al. (2018), Piech et al. (ICML'2015) and Bastings et al. (EMNLP'2017). We are not aware of any other work (except for baselines that we already used) that targets a task of code-to-sequence and is reproducible. We will make all of our code publicly available and reusable such that future works will have concrete and reproducible baselines. Our datasets are already publicly available and used in works such as [a].

[a] Anonymous, Structured Neural Summarization, Under review for ICLR'19 - https://openreview.net/forum?id=H1ersoRqtm

---

### Meta-Review · Area_Chair1 · 2018-12-13
**Marginally novel method, extensive experiments, reasons for experimental results not extremely clear.**

**Confidence:** 4
**Recommendation:** Accept (Poster)

**Metareview:**

Overall this paper presents a few improvements over the code2vec model of Alon et al., applying it to seq2seq tasks. The empirical results are very good, and there is fairly extensive experimentation.

This is a relatively crowded space, so there are a few natural baselines that were not compared to, but I don't think that comparison to every single baseline is warranted or necessary, and the authors have done an admirable job. One thing that still is quite puzzling is the strength of the "AST nodes only baseline", which the authors have given a few explanations for (using nodes helps focus on variables, and also there is an effect of combining together things that are close together in the AST tree). Still, this result doesn't seem to mesh with the overall story of the paper all that well, and again opens up some obvious questions such as whether a Transformer model trained on only AST nodes would have done similarly, and if not why not.

This paper is very much on the borderline, so if there is space in the conference I think it would be a reasonable addition, but there could also be an argument made that the paper would be stronger in a re-submission where the above questions are answered.